# Substrate Structured Bournonite CuPbSbS₃ Thin Film Solar Cells

Limei Lin [1,2,*], Rui Da [1,2], Chenqi Zheng [1,2], Ruibo Zeng [1,2] and Junda Ding [1,2]

1. Fujian Provincial Engineering Technology Research Center of Solar Energy Conversion and Energy Storage, College of Physics and Energy, Fujian Normal University, Fuzhou 350117, China; darui2662@gmail.com (R.D.); zhengchenqi2@gmail.com (C.Z.); zengruibo265@gmail.com (R.Z.); jundading155@gmail.com (J.D.)
2. Fujian Provincial Collaborative Innovation Center for Advanced High-Field Superconducting Materials and Engineering, Fuzhou 350117, China
* Correspondence: linlm@fjnu.edu.cn

**Abstract:** CuPbSbS₃ has excellent photoelectric properties, such as high element abundance and optical absorption coefficient, and a suitable band gap, which is a material with the potential for absorbing layers of high-quality thin film solar cells. In addition, CuPbSbS₃ is a material with a three-dimensional structure, which can guide the carrier to transport in all directions, so its performance can be regulated in multiple dimensions. At present, the substrate structure is often used in efficient solar cells since this structure does not affect other functional layers when the absorption layer is subjected to harsh annealing conditions. However, there have been no reports of the substrate structure of CuPbSbS₃ solar cells so far. Therefore, in this work, CuPbSbS₃ films deposited on a stable substrate of molybdenum (Mo) were prepared with butyldithiocarbamic acid (BDCA) solution, and the preparation process of reaction mechanism was described in detail. It was found that the band gap of the CuPbSbS₃ thin film was 2.0 eV and the absorption coefficient was up to $10^5$ cm$^{-1}$, which is expected to be applied to the top absorption layer material in laminated cells. Thus, we first built a Glass/Mo/CuPbSbS₃/CdS/ZnO/ITO substrate structured solar cell. From this, a photoelectric conversion efficiency of 0.094% was achieved. This work provides a tentative exploration for the future development of substrate structured CuPbSbS₃ solar cells.

**Keywords:** CuPbSbS₃; BDCA solution; solar cell; substrate structure





## 1. Introduction

Thin film solar cells have gained extensive attention owing to their advantages such as their high efficiency, low cost, and flexibility. In order to further reduce the price of thin film solar cells and improve their application value, absorbing layer materials as Cu₂ZnSnS₄ (CZTS), Sb₂S₃, CuSbS₂, and CuPbSbS₃, which have abundant element reserves and good photoelectric performance, have attracted more and more attention from researchers [1]. In CZTS thin film, multiple defects are easy to form, and the secondary phases of N-type semiconductors such as ZnS and SnS₂ make it easy to affect the device's efficiency [2,3]. Sb₂S₃ is a quasi-one-dimensional structural material, and its crystal orientation is complex and difficult to adjust, which will seriously affect the charge transfer efficiency [4]. CuSbS₂ is a two-dimensional material, and the 2D monolayered crystal structure prevents the overlap of the electron clouds between the layers, allowing efficient transportation of the carriers in an intra-layer manner but blocking their transportation along the out-of-plane direction [5]. Recently, some pioneering works have shown that the introduction of Pb source into CuSbS₂ to form CuPbSbS₃ can improve the two-dimensional crystal structure to a three-dimensional crystal. The three-dimensional electron cloud structure will help improve the charge transfer efficiency, which is expected to demonstrate advanced electronic dimensionality [6].

The first appearance of $CuPbSbS_3$ in solar cells was a top structure device of Glass/ITO/CdS/$CuPbSbS_3$/Spiro-OMeTAD/Au prepared by Professor Tang's group using the spin-coating method. This device achieved a photoelectric conversion efficiency (PCE) of 2.23% and an open voltage ($V_{OC}$) of 699 mV [6]. Subsequently, they further updated the PCE to 2.65% in 2021 by optimizing the crystallinity of the $CuPbSbS_3$ film, which is also the champion device efficiency of $CuPbSbS_3$ so far [7].

The preparation method of $CuPbSbS_3$ and its application in solar cells are relatively not enough. The common preparation method for similar four-component compound CZTS films is the solution-spin coating method, which can easily control the content of each component and form a film with uniform texture and controllable thickness. However, since the $K_{sp}$ of PbS, $Sb_2S_3$, and $Cu_2S$ are too different with respect to orders of magnitude, it is difficult to prepare $CuPbSbS_3$ with an aqueous solution-spin coating method. Due to the large number of coordination bonds to be paired on the S atom in butyldithiocarbamic acid (BDCA) solution, it can chelate metal cations very effectively and form a stable gel, finally precipitate the corresponding metal sulfides through high temperature heating [8,9]. The above BDCA solution method has achieved great success in the preparation of other photovoltaic absorption materials and its feasibility has been fully verified [8,10]. In this regard, we tried to use BDCA solution to prepare the precursor solution. A $CuPbSbS_3$ film can be formed under the joint reaction of a variety of metal sulfides [9]. However, in this process, the $CuPbSbS_3$ film will inevitably undergo relatively complex annealing conditions, which is easy to cause irreversible effects on other functional layers (especially in the top structure solar cell). Consequently, we use chemically stable, electrically conductive, and single-crystal oriented Mo films as substrates, which enables $CuPbSbS_3$ deposited on Mo substrates to perform more demanding process operations, and also benefits the substrate structure design of $CuPbSbS_3$ solar cells.

In this article, we have prepared a pure p-type $CuPbSbS_3$ thin film with wide bandgap of 2.0 eV and high absorption coefficient of $\sim 10^5$ $cm^{-1}$. The characteristics make it a top cell material in laminated cells. In order to further explore its potential of photoelectric device, we have assembled a substrate structured $CuPbSbS_3$ solar cells for the first time. The final assembled device has a photoelectric conversion efficiency of 0.094%. Lastly, we analyzed the adverse factors of the cell performance from the band structure and conductive properties.

## 2. Experimental Details

### 2.1. Device Fabrication Process

The illustrative diagram of substrate structure $CuPbSbS_3$ thin films solar cells can be seen in Figure 1. Firstly, Mo thin film was deposited by direct current magnetron sputtering. To increase the adhesion of the film to the glass substrate, a double-Mo-film with a ~300 nm higher power (150 W) layer and a ~800 nm top lower power (100 W) layer was prepared.

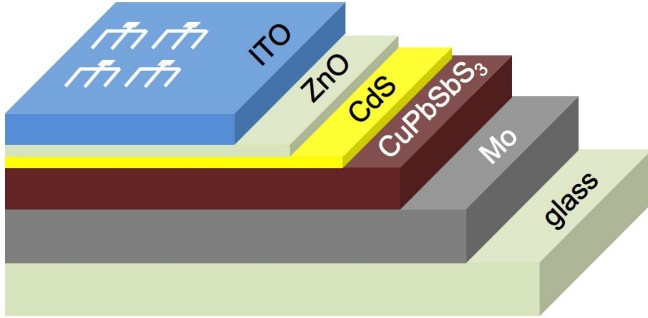

**Figure 1.** The diagram of substrate structure $CuPbSbS_3$ thin film solar cells.

Secondly, a $CuPbSbS_3$ thin film was prepared by spin coating on the Mo substrate. First of all, the Cu, Pb and Sb precursor solutions were prepared by the BDCA solution method.

A total of 5 mL of $CS_2$ (Aladdin, >99.9%) and 10 mL of ethanol (Huachen, >99.7%) were mixed in a 40 mL flask under high-speed magnetic stirring at room temperature. 0.5724 g (4 mmol) of $Cu_2O$ (Aladdin, 99%) was subsequently loaded into the flask. Afterwards, 8 mL of n-butylamine (Aladdin, ≥99.5%) was added dropwise into the mixed solution, followed by a further stirring for 10 min at least to obtain a clear Cu precursor solution. The Pb and Sb precursor solutions were prepared with similar procedure, which contain 0.6696 g (3 mmol) of PbO (Aladdin, 99.9%) and 1.166 g (4 mmol) of $Sb_2O_3$ (Aladdin, 99.5%), respectively. Subsequently, three solutions were mixed together with a mole ratio of 0.9:1:1 (Cu:Pb:Sb) under magnetic stirring for 30 min to form a light-yellow solution. Next, the $CuPbSbS_3$ precursor film was spun onto the Mo substrate at 800 rpm for 10 s and 2500 rpm for 60 s. The film was then placed on a 70 °C hot plate for 5 min, afterwards a 275 °C hot plate for 5 min, subsequently cooled down to room temperature. The thickness of thin film was controlled by repeating spin-coating/sintering process. Continuing three cycles yielded a ~500 nm $CuPbSbS_3$ thin film. The solution preparation and thermal annealing process were carried out inside a $N_2$ filled glove box.

Finally, photovoltaic devices were completed based on the aforementioned $CuPbSbS_3$ thin films as follows: chemical bath deposition of CdS (~30 nm), sputtering of ZnO (~30 nm) and ITO (~200 nm), and thermal evaporation of aluminum grids. Devices were then scribed into 0.163 $cm^2$.

### 2.2. Measurement and Characterization

The structure and phase properties of the thin film were studied by X-ray diffractometer (XRD, Rigaku, Ultima IV, Japan) with Cu $K_\alpha$ irradiation ($\lambda$ = 1.5406 Å) and Raman spectrometer (Horiba, LABRAM-HR, Japan) with laser excitation at 532 nm. The morphology and the composition of the film were characterized by field emission scanning electron microscope equipped with an energy dispersive spectrometer (FESEM, Hitachi, SU-8010, Japan). The thermal stability of the film was characterized by differential scanning calorimetry (DSC, STA449E3, STA449E3, Germany) and thermogravimetric analysis (TGA, Netzsch, QMS403D, Germany).Ultraviolet photoelectron spectroscopy (UPS, Thermo, ES-CALAB 250XI, USA) was used to investigate valence band maximum (VBM) and Fermi energy of film by using a He I (21.2 eV) gas discharge lamp with −5 V sample bias. A UV-Vis-NIR spectrophotometer (Perkin Elmer, Lambda 950, USA) was used to test the transmission spectrum of the $CuPbSbS_3$ thin film prepared on glass. The current density-voltage (J-V) was measured by a Keithley 2400 Source Meter SMU Instruments and an AAA solar simulator (SAN-EI Electric, XES-40S1, Japan). The external quantum efficiency (EQE) of the devices was performed on a solar cells quantum efficiency measurement system (PV Measurements, QEX10, USA). The impedance spectra were measured from 1 Hz to 1 MHz via an electrochemical workstation (Zahner, Zennium, Germany) in the dark.

### 3. Results and Discussion

BDCA is easily synthesized by the combination reaction of n-butylamine ($CH_3(CH_2)_3NH_2$) and carbon disulfide ($CS_2$), and used to dissolve a variety of metal oxides (MO) to form sulfur-based organometallic complex precursor solutions (reaction 1). The organometallic complex will be pyrolyzed to form the sulfur-based complex (reaction 2).

$$CH_3(CH_2)_3NH_2 \xrightarrow{CS_2} CH_3(CH_2)_3HNS_2H \xrightarrow{MO} M(S_2CNHC_4H_9)_x \tag{1}$$

$$M(S_2CNHC_4H_9)_x \rightarrow MS_x + x(S_2CNHC_4H_9) \tag{2}$$

The Cu-Pb-Sb precursors are subsequently mixed at a mole ratio of Cu:Pb:Sb = 0.9:1:1. In order to explore the formation process of substances and afterwards determine the drying and annealing temperatures of the sintering procedure, TGA and DSC of Cu-Pb-Sb mixed solution was performed. Figure 2a shows the phase change process of the Cu-Pb-Sb mixed solution. The endothermic peak at 70 °C attributes to the evaporation of ethanol. A

broad endothermic peak around 125 °C is associated to removal of free BDCA. At 230 °C, the Cu-Pb-Sb mixed solution has the greatest mass loss, owing to the fact that the BDCA solution begins to dissociate the metal sulfides and starts the reaction to form $CuPbSbS_3$. In order to confirm this hypothesis, XRD tests were conducted on the as-spin coating film annealing at different temperatures. As shown in Figure 2b, the film shows amorphous properties after annealing at 70 °C, in which the diffraction peak at 36.41° belongs to $Cu_2O$ preferentially precipitated in solution. Due to the small charge and coordination number of $Cu^+$, its polarization ability is weak, and the coordination bond strength of the coordination compound formed with BDCA solution is also low. Therefore, the precipitation of $Cu_2O$ was observed at 70 °C. The phenomenon of producing a p-type light-absorbing material of copper oxide in the film, which maybe helps to promote the absorption of sunlight, and is also easy to form a copper poor $CuPbSbS_3$ thin film benefit to improve the device efficiency similar in copper based solar cells, such as $Cu(In,Ga)(S,Se)_2$ and CZTS. Above situation is consistent with the purpose of the proportion of copper poor elements used in the design of the experiment. The XRD pattern of the sample annealing at 130 °C is consistent with that of 70 °C, which indicates that $CuPbSbS_3$ cannot be generated in the sample below 125 °C. However, when the temperature was increased to 230 °C, the sample showed obvious multiple diffraction peaks, which belonged to $CuPbSbS_3$ phase (PDF#76-1999). The Cu-Pb-Sb mixed sulfides dissociates and stable $CuPbSbS_3$ phase is formed around 230 °C. Based on the above XRD and TG analysis, we can obtain a more suitable preparation process, and set the drying and annealing temperatures of the final film preparation to 70 °C and 275 °C accordingly.

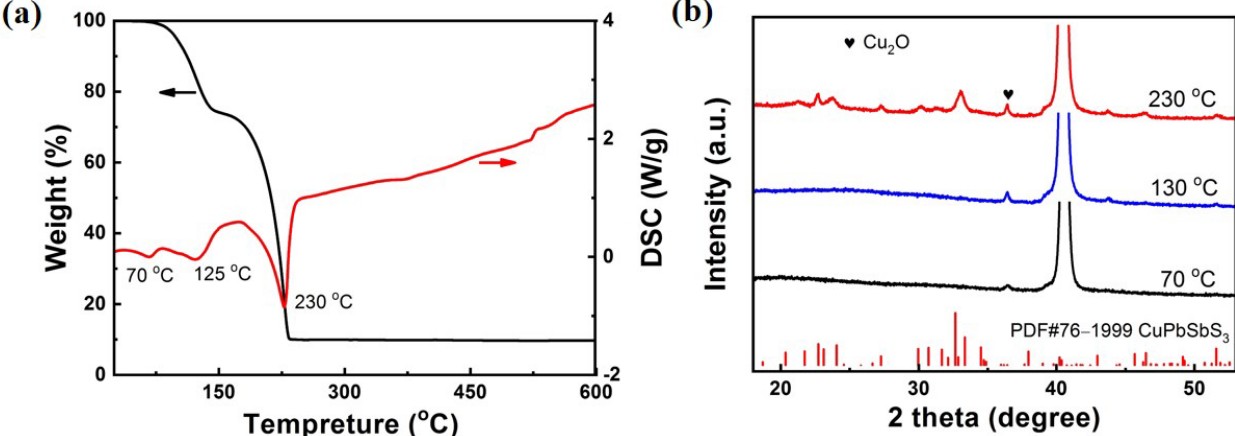

**Figure 2.** (**a**) TGA curves of Cu-Pb-Sb mixed solution, (**b**) XRD results of $CuPbSbS_3$ films.

Generally a ~500 nm absorber film is beneficial to photoelectric performance of thin film solar cells [2]. The thickness of thin film can be easily adjusted by repeating spin-coating/sintering process. Thus, $CuPbSbS_3$ films were prepared by repeating spin-coating/sintering process. To clarify the formation process of $CuPbSbS_3$ films by the spin coating method, the morphology, composition, and structural characteristics of each layer were investigated. As shown in Figure 3, the 1 layer (1L) film is a uniform film with a thickness of about 220 nm, and the grain size is about 60 nm. However, there are some ~100 nm special particles on its surface. After repeating the spin-coating/sintering process, the prepared 2 layers (2L) and 3 layers (3L) films are uniform and dense without the special large particles. The grain sizes of 2L and 3L films are similar to that of 1L film. The thicknesses of 2L and 3L film are 400 nm and 560 nm, respectively.

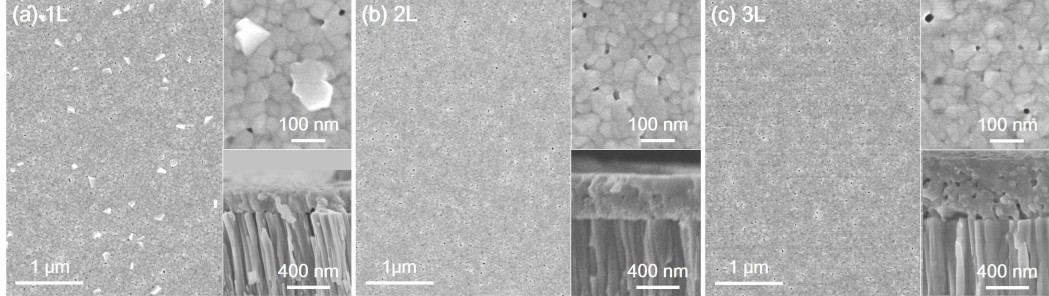

**Figure 3.** Surface and cross-section SEM images of (**a**) 1 layer (1L), (**b**) 2 layers (2L) and (**c**) 3 layers (3L) CuPbSbS$_3$ films.

The energy dispersive spectrometer (EDS) was used to explore the component of 1L, 2L and 3L films. EDS tests were applied at five different points for uniformity test and 100 μm$^2$ area for average value test. SEM images of CuPbSbS$_3$ films for the EDS test are presented in Figure S1 of Supplementary information and the results are summarized in Table 1. The EDS results of 1L sample shows that the average Cu:Pb:Sb:S ratio of 100 μm$^2$ area is 1.0:1.4:0.99:4.1, which is slightly different from the 1.0:1.0:1.0:3.0 of chemometric CuPbSbS$_3$, mainly displaying a Pb-rich and S-rich constituent. We hypothesize that some PbS failed to participate in the reaction to form CuPbSbS$_3$ while PbS is finally precipitated in BDCA solution. This phenomenon is confirmed in the XRD pattern (Figure 4a), and the diffraction peaks of PbS are detected in all of the XRD patterns of the films. However, the average element content detected at the positions of large particles of 1L film (point1, 2 and 3) is 1.0:1.5:1.4:4.0, which is different from the average element content of the 100 μm$^2$ area. On account of the higher Sb and S contents, it can be inferred that the large particles on 1L film maybe Sb$_2$S$_3$ grains which was found similarly in the literature [11]. For the 2L and 3L films, the element content ratios at each position are relatively consistent, which indicates that the prepared films are high uniformity and that the element ratios are close to the chemometric CuPbSbS$_3$.

**Table 1.** EDS results of CuPbSbS$_3$ films.

| Cu:Pb:Sb:S | 1L | 2L | 3L |
|---|---|---|---|
| Point 1 | 1.0:1.5:1.4:3.2 | 1.0:1.2:0.91:3.0 | 1.0:1.1:0.94:2.6 |
| Point 2 | 1.0:1.6:1.5:5.1 | 1.0:1.2:0.92:2.7 | 1.0:1.2:1.0:2.6 |
| Point 3 | 1.0:1.4:1.3:3.8 | 1.0:1.3:1.0:3.0 | 1.0:1.2:1.0:2.5 |
| Point 4 | 1.0:1.5:0.97:4.2 | 1.0:1.1:0.89:2.6 | 1.0:1.1:0.94:2.5 |
| Point 5 | 1.0:1.3:0.89:5.0 | 1.0:1.3:1.0:3.5 | 1.0:1.2:1.0:2.7 |
| Average of points | 1.0:1.5:1.4:4.0 (P1-3)<br>1.0:1.4:0.93:4.6 (P4-5) | 1.0:1.2:0.95:3.0 | 1.0:1.2:0.98:2.6 |
| Average of 100 μm$^2$ area | 1.0:1.4:0.99:4.1 | 1.0:1.2:0.96:2.8 | 1.0:1.2:1.0:2.9 |

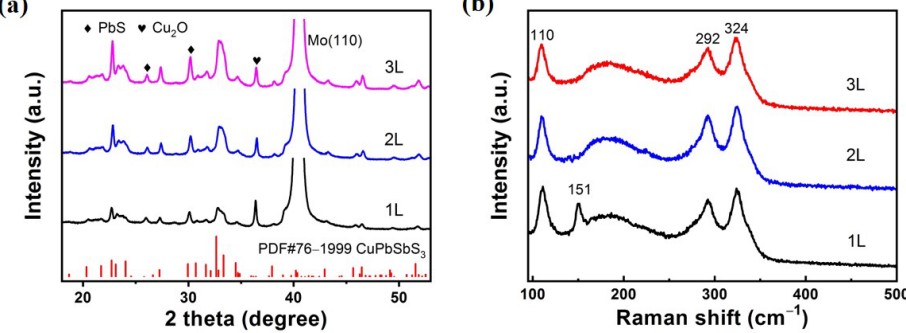

**Figure 4.** (**a**) XRD patterns and (**b**) Raman spectra of CuPbSbS$_3$ films.

In Figure 4a, the XRD patterns show that the CuPbSbS$_3$ diffraction peaks of the 2L and 3L films are obviously enhanced and the crystallinity is improved after repeated spin-coating and annealing. With the increase of spin-coating times, Cu$_2$O and PbS phases still exist, which is consistent with the above conclusion. Since the solubility of Cu$_2$O in BDCA is not high, it is precipitated easily even though it is annealed at low temperature. PbS is precipitated when BDCA is dissociated at a high temperature. However, there is no diffraction peaks of Sb compound can be detected even in 1L sample, which proves that the content of Sb$_2$S$_3$ particles is low. Moreover, it is also found from SEM images that the Sb$_2$S$_3$ particles only appear on 1L film, meaning little content of Sb$_2$S$_3$ secondaryphase in the film.

To further verify the above results, we conduct Raman tests and show them in Figure 4b. Raman spectra show that the 1L film annealed at 270 °C has obvious Raman peaks of CuPbSbS$_3$ phase (110 cm$^{-1}$, 292 cm$^{-1}$ and 342 cm$^{-1}$), which is consistent with the literature [12]. A 151 cm$^{-1}$ peak referred to the Sb$_2$S$_3$ phase only appears on 1L film rather than 2L and 3L films, which verifies again the existence of Sb$_2$S$_3$ on 1L film except other samples [13]. These results are in agreement with those of SEM and XRD.

To sum up, we can clearly understand the formation process of CuPbSbS$_3$ film. Firstly, Cu$_2$O is precipitated easily due to volatilization of BDCA solution even at low temperature of 70 °C. Secondly, Sb$_2$S$_3$ and PbS are precipitated when 1L film is spin-coated, but only PbS phase is precipitated in 2L and 3L samples. Finally, a main phase is CuPbSbS$_3$ by annealing above 230 °C. Although the composition of the film is complex, the tiny secondary phases Cu$_2$O, Sb$_2$S$_3$ and PbS are p-type semiconductors and light-absorbing materials similar to CuPbSbS$_3$. Therefore, the existence of these secondary phases may also bring certain performance improvements to solar cells.

In order to explore the potential of CuPbSbS$_3$ films as absorption materials, we tested the transmission spectrum and calculated absorption coefficient of a 3L-CuPbSbS$_3$ film on glass, as shown in Figure 5. The absorption coefficient of CuPbSbS$_3$ film can reach up to 10$^5$ cm$^{-1}$, which fully proves its potential as an absorption material for solar cells. The calculated band gap of CuPbSbS$_3$ is 2.0 eV which is different from its theoretical direct band gap of 1.41 eV [14]. In fact, due to differences in preparation methods, the actual band gap value reported now will be slightly different. For example, Richard L. Brutchey's team prepared CuPbSbS$_3$ with a band gap of 1.21 eV in 2020 [15]. In the same year, the band gap of CuPbSbS$_3$ prepared by J. Tang's team is 1.31 eV [6]. David J. Lewis's group prepared CuPbSbS$_3$ powder with a band gap of 1.55 eV in 2021 [12]. In the process of preparing CuPbSbS$_3$, secondary phases such as Cu$_2$O (2.2–2.48 eV), PbS (0.76–1.91 eV) and Sb$_2$S$_3$ (1.7–1.8 eV) are easily generated, and the formation of these secondary phases will have a certain impact on the final band gap of the film [16–20]. The wide band gap value of 2.0 eV indicates that CuPbSbS$_3$ can be applied to the laminated top solar cell material, which helps to improve the light absorption of the laminated solar cell in the short wavelength segment. So, we prepare CuPbSbS$_3$ films with repeating the spin coating/sintering process three times, and assemble them into Glass/Mo/CuPbSbS$_3$/CdS/ZnO/ITO structure of the solar cell to probe into its development potential in laminated top solar cells.

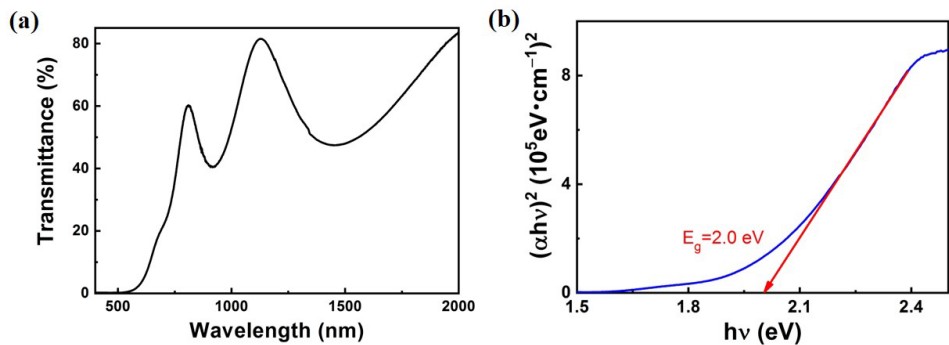

**Figure 5.** (**a**) transmittance and (**b**) absorption coefficient ($\alpha$) of CuPbSbS$_3$ film.

The photovoltaic performance of the device is shown in Figure 6a. The obtained open voltage ($V_{OC}$), short circuit current density ($J_{SC}$), filling factor (FF) and PCE of $CuPbSbS_3$ cell are 309 mV, 0.91 mA/cm$^2$, 33.6% and 0.094%, respectively. The collection efficiency of photoexcited carriers was tested from the EQE spectrum presented in Figure S3. The integrating current obtained with EQE spectrum is 0.81 mA/cm$^2$ which is consistent with the $J_{SC}$ obtained from J-V curve. To investigate the charge extraction and recombination processes in the solar device, the electrochemical impedance spectroscopy (EIS) experiment was executed under dark conditions, which is displayed in Figure S4; the corresponding equivalent circuit model is also illustrated. The fitted equivalent circuit model is composed of series resistance ($R_s$), charge transfer resistance ($R_{tr}$), and recombination resistance ($R_{rec}$), as well as parallel capacitors of $C_{tr}$ and $C_{rec}$. The fitted $R_s$, $R_{tr}$ and $R_{rec}$ are 36 ohm, 283 ohm and 481 ohm, respectively. The $R_{rec}$ is inversely proportional to the charge recombination rate of the device. $R_{rec}$ of the $CuPbSbS_3$ solar cell is lower than that of the emerging CZTS and $Sb_2S_3$ solar cells by one order of magnitude, meaning a more recombination loss in our device [2,21]. The dark J-V curve of the $CuPbSbS_3$ device is used to determine the diode properties. As shown in Figure S5, the reverse saturation current ($J_0$) is $1.74 \times 10^{-4}$ mA/cm$^2$, which is about two orders of magnitude larger than that of the emerging CZTS and $Sb_2S_3$ solar cells, indicating that there is an excess of short-circuit channels in the $CuPbSbS_3$ device [2,21]. The existent of a lot of short-circuit channels also leads to poor short circuit current density and open voltage.

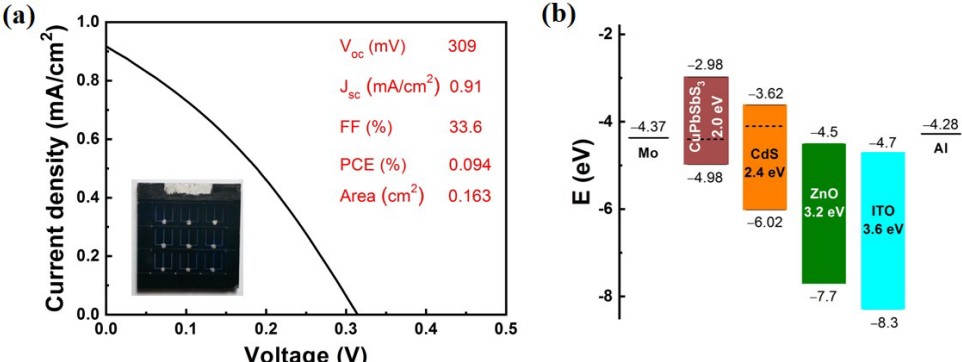

**Figure 6.** (**a**) J-V and (**b**) band scheme of $CuPbSbS_3$ thin film solar cell.

To explore the mechanism factors affecting solar cell performance, the band scheme of the $CuPbSbS_3$ device must be observed. The conduction band minimum (CBM), valence band maximum (VBM), and Fermi level ($E_f$) of $CuPbSbS_3$ and CdS films were measured by ultraviolet electron spectroscopy (UPS), and the parameters of other layers were taken from the reference [22]. The energy level structure of $CuPbSbS_3$ thin-film solar cells was obtained (Figure 6b). It can be seen that the Fermi energy level is 0.58 eV above VBM of $CuPbSbS_3$ film, which means serious deep-level defects in the film and leads to poor conductivity and serious carrier recombination, resulting in small short-circuit current density. At the same time, the Fermi level difference between $CuPbSbS_3$ and CdS is only about 0.3 eV, and then causes the low open-circuit voltage. It can also be seen from the band scheme that the front and back electrodes (Mo and Al) are not very suitable electrode materials. The work function of Mo is 4.3 eV and VBM of $CuPbSbS_3$ film is 4.98 eV, which induces a big cliff for hold transport and will cause serious interface carrier recombination. To test the contact between the $CuPbSbS_3$ film and the Mo substrate, the I-V properties of Mo and $CuPbSbS_3$/Mo structures were tested. As shown in Figure S6, there is a non-ohmic contact between $CuPbSbS_3$ and Mo due to a potential barrier between them which will seriously affect the performance of the solar cell. In brief, reducing the serious carrier recombination during $CuPbSbS_3$ or interfaces is a key factor to improve the efficiency of the $CuPbSbS_3$ thin film solar cells. In order to improve the carrier transportation and collection efficiencies, $CuPbSbS_3$ thin films with better crystallinity and conductivity and fewer defects should be

prepared. An interface modification between Mo and $CuPbSbS_3$ is needed to reduce the hole transport barrier. Other strategies to promote the device performance by enhancing the absorption efficiency (including application of photon convertors such as quantum dots, plasmonic metallic nano-components, or a bottom metallic mirror) have proved to be feasible both in theory and via experiments [23].

### 4. Conclusions

Precisely controlling the thickness of the film and its element gradient content is the key to the preparation of the quaternary compound $CuPbSbS_3$, so we used a BDCA solution/multiple spin-coating method to prepare the structured bournonite $CuPbSbS_3$ thin films. At the same time, $CuPbSbS_3$ was deposited on Mo substrate to prepare the device with the substrate structure, which is beneficial to perform more demanding process operations on $CuPbSbS_3$ than the top structure devices. In the preparation process, secondary phases of $Cu_2O$, PbS and $Sb_2S_3$ appear. The emergence of these p-type semiconductors results in a band gap of 2.0 eV in $CuPbSbS_3$, which can be applied to top cell materials in laminated solar cells. Finally, the $CuPbSbS_3$ film as an absorption layer was assembled into the Glass/Mo/$CuPbSbS_3$/CdS/ZnO/ITO substrate structured solar cell components, which got 0.094% of the photoelectric conversion efficiency. The deep-level defects of the $CuPbSbS_3$ film and mismatching of the energy level structure of $CuPbSbS_3$/CdS heterojunction are essential reasons for the poor performance. The high energy barrier and the non-ohmic contact are shown in the band structure and electrical level contact, which will have a fatal effect on the carrier collection efficiency. This work is the first to apply $CuPbSbS_3$ to solar cells with a substrate structure, and provides a certain guiding significance for the preparation of $CuPbSbS_3$ materials and the development of $CuPbSbS_3$ solar cells in the future.

**Supplementary Materials:** The following supporting information can be downloaded at: https://www.mdpi.com/article/10.3390/cryst13081256/s1. Figure S1. SEM images of $CuPbSbS_3$ films for EDS test: (a) 1 layer (1L), (b) 2 layers (2L) and (c) 3 layers (3L). Figure S2. UPS results of (a) $CuPbSbS_3$ and (b) CdS thin films. Figure S3. EQE spectrum of $CuPbSbS_3$ thin film solar cell. Figure S4. Electrochemical impedance spectroscopy measurement of $CuPbSbS_3$ thin film solar cell (inset equivalent circuit diagram). Figure S5. Dark J-V characteristic of $CuPbSbS_3$ thin film solar cell. Figure S6. I-V Curves of (a) Mo and (b) $CuPbSbS_3$/Mo structures (inset resistance test modes).

**Author Contributions:** Conceptualization, L.L.; methodology, L.L. and R.D.; validation, R.D., C.Z. and R.Z.; investigation, C.Z., R.Z. and J.D.; data curation, L.L. and R.D.; writing—original draft preparation, L.L. and R.D.; writing—review and editing, L.L.; funding acquisition, L.L. All authors have read and agreed to the published version of the manuscript.

**Funding:** This work was supported by the National Natural Science Foundation of China (Grant No. 61974028; 62204041).

**Data Availability Statement:** The data presented in this study are available on request from the corresponding author.

**Conflicts of Interest:** The authors declare no competing financial interests.

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
