# Peer review of "Substrate Structured Bournonite CuPbSbS3 Thin Film Solar Cells"

_crystals, doi:10.3390/cryst13081256_

Round 1

Reviewer 1 Report

Comments and Suggestions for Authors

The submisson is a report of  experimental preparation of some multilayr photovoltaic structure including absorption layer of CuPbSeS_3. The descriptions of a structure manufacturing and of measurement are relatively interesting and desrve probably publication but some revision of the paper is required.

In abstract ‘band gap’ rather than ‘optical band gap’ should be used (the same in the text).

In abstract ‘BDCA’ must be developed, ‘Mo’ and ‘FTO’ as well.

In abstract, ‘absorption coefficient 1 cm-1’ must be explained, if is correct.

In conclusions the phrase in lines 281-3 is completely unclear – must be clarified.

The obtained efficiency 0.094 % is extremely poor and authors shoould rather discuss ways how to improve the efficiency and what is the reason of so low achievement. It is obvious that the limiting factor in thin films in application to absorbing layers is the short path for photons and this should be discussed in the paper. To enhance absorption efficiency some simple methods should be applied – inluding application of photon convertors like quantum dots or plasmonic metallic nano-components. The latter are especially effective in thin layer absobers as they operate within a  near-field zone of plasmons in nanoparticle metallic concentrators deposited on the top of a thin layer. The effect is very large and is of quantum character related to coupling of plasmons with electron band system in a semiconductor substrate. The theory is formulated and repeated in many papers and described in details in monograph “Quantum Nano-Plasmonics” by W. Jacak (Cambridge University Press 2020) where the review of related experimental achivement has been presented as well. How to apply plasmonic converters in thin layer substrates cf. also Nanomaterials 2019, 9, 3; doi:10.3390/nano9010003.  It will be beneficial to add the related comment in conclusion, especially in view of a poor efficiency obtained, rather not optmistic for photovoltaic applications. To enhance the photon path length and increase absorption rate, application of a bottom metallic mirrow would be also convenient.

Units should be separated by a space from the text – e.g., in line 80. The references should be cited rather before punctation signs with preserving a space as well.

In bibliography, what does mean [J] added to items?

The entire text requires thorough proofreading and linguistic corrections.

The submisson is a report of  experimental preparation of some multilayr photovoltaic structure including absorption layer of CuPbSeS_3. The descriptions of a structure manufacturing and of measurement are relatively interesting and desrve probably publication but some revision of the paper is required.

In abstract ‘band gap’ rather than ‘optical band gap’ should be used (the same in the text).

In abstract ‘BDCA’ must be developed, ‘Mo’ and ‘FTO’ as well.

In abstract, ‘absorption coefficient 1 cm-1’ must be explained, if is correct.

In conclusions the phrase in lines 281-3 is completely unclear – must be clarified.

The obtained efficiency 0.094 % is extremely poor and authors shoould rather discuss ways how to improve the efficiency and what is the reason of so low achievement. It is obvious that the limiting factor in thin films in application to absorbing layers is the short path for photons and this should be discussed in the paper. To enhance absorption efficiency some simple methods should be applied – inluding application of photon convertors like quantum dots or plasmonic metallic nano-components. The latter are especially effective in thin layer absobers as they operate within a  near-field zone of plasmons in nanoparticle metallic concentrators deposited on the top of a thin layer. The effect is very large and is of quantum character related to coupling of plasmons with electron band system in a semiconductor substrate. The theory is formulated and repeated in many papers and described in details in monograph “Quantum Nano-Plasmonics” by W. Jacak (Cambridge University Press 2020) where the review of related experimental achivement has been presented as well. How to apply plasmonic converters in thin layer substrates cf. also Nanomaterials 2019, 9, 3; doi:10.3390/nano9010003.  It will be beneficial to add the related comment in conclusion, especially in view of a poor efficiency obtained, rather not optmistic for photovoltaic applications. To enhance the photon path length and increase absorption rate, application of a bottom metallic mirrow would be also convenient.

Units should be separated by a space from the text – e.g., in line 80. The references should be cited rather before punctation signs with preserving a space as well.

In bibliography, what does mean [J] added to items?

The entire text requires thorough proofreading and linguistic corrections.

Comments on the Quality of English Language

The entire text requires thorough proofreading and linguistic corrections.

English phraseology needs to be improved in many places. 

Reviewer 2 Report

Comments and Suggestions for Authors

Which is p-type and n-type material in the device>

What is the role of CdS and Mo?

Work function of material to be discussed

Where is the dark current of the device?

Impedance to be performed and discussed.

IPCE to be measured and it is compared with the current density.

Voltage and current are very low. In what way this work is new or novel.

Compare the results with the other thin film solar cells

Comments on the Quality of English Language

Ok

Round 2

Reviewer 1 Report

Comments and Suggestions for Authors

The authors made some improvement of the presentation  but some further correction seems to be needed.

The sentence in line 189-190 must be clarified (rewritten with proper grammar).

Quality of graphics in Fig. 6, 7 must be improved to be readable.

In Conclusion it must be commented why the efficiency of the constructed cell is so low. Authors suggeted in the revised text that the hampering factor is a hight recombination of excitons. Please address to other possible factors reducing efficiency of the construction (perhaps also inappropriate adjustment of other layers in the entire structure for the optimization of operation). 

The operation of this particular cell should be briefly commented on in the text beforehand to identify the critical points of the multilayer design.

Comments on the Quality of English Language

linguistic proofreading is required

using of some phrases should verified 

(e.g., 'newcomer' in abstract) 

Author Response

Manuscript ID: Crystals-2503903

Paper title: Substrate structured bournonite CuPbSbS3 thin films solar cells

Authors: Limei Lin *, Rui Da, Chenqi Zheng, Ruibo Zeng and Junda Ding

Thanks very much for your letter and useful comments. We have studied these comments carefully and modified accordingly in the revision. The main corrections are marked in red in the revision and the responds to the reviewer’s comments are summarized as follows:

Reviewer 1 Comments:

The authors made some improvement of the presentation but some further correction seems to be needed.

° The sentence in line 189-190 must be clarified (rewritten with proper grammar).

Thanks for your careful comment. The grammar mistake is corrected in line 179 of the revision, as follows: “On account of the higher Sb and S contents, it is inferred that the large particles on 1L film maybe Sb2S3 grains which was found similarly in the literature [11].”

° Quality of graphics in Fig. 6, 7 must be improved to be readable.

Thanks for your useful advices. We have redrawn all of the graphics for more clarity and rational layout, such as combine Figures 4 and 5 to form Figure 4, and the subsequent figure labels are revised. Figures 6 and 7 are improved and shown in Figures 5 and 6, respectively. Meanwhile Figure S6 is also adjusted accordingly.

Figure 5 (a) transmittance and (b) absorption coefficient (α) of CuPbSbS3 film.

Figure 6 (a) J-V and (b) band scheme of CuPbSbS3 thin film solar cell.

Figure S6 I-V Curves of (a) Mo and (b) CuPbSbS3/Mo structures (inset resistance test modes).

° In Conclusion it must be commented why the efficiency of the constructed cell is so low. Authors suggested in the revised text that the hampering factor is a high recombination of excitons. Please address to other possible factors reducing efficiency of the construction (perhaps also inappropriate adjustment of other layers in the entire structure for the optimization of operation).

Thank you very much for your advices. In conclusion, we added some details for clarifying the poor efficiency. As follows: The deep-level defects of the CuPbSbS3 film and mismatching of the energy level structure of CuPbSbS3/CdS heterojunction are essential reasons for the poor performance. In the line 266-269, we also appended some advices to improve the cell performance: In order to improve the carrier transportation and collection efficiencies, CuPbSbS3 thin films with better crystallinity and conductivity and fewer defects should be prepared. An interface modification between Mo and CuPbSbS3 is needed to reduce the hole transport barrier.

° The operation of this particular cell should be briefly commented on in the text beforehand to identify the critical points of the multilayer design.

Thanks for your advices. In the revision, we have illustrate the issues of the multilayer design. As shown in line 153-157: Generally a ~500 nm absorber film is beneficial to photoelectric performance of thin film solar cells [2]. The thickness of thin film can be easily adjusted by repeating spin-coating/sintering process. Thus, CuPbSbS3 films were prepared by repeating spin-coating/sintering process. To clarify the formation process of CuPbSbS3 films by the spin coating method, the morphology, composition and structural characteristics of each layer were investigated.

° Comments on the Quality of English Language, linguistic proofreading is required. Using of some phrases should verified (e.g., 'newcomer' in abstract)

Thanks for your careful comments. We reread thoroughly the whole article and corrected its mistakes. Modified details as follows:

Abstract: “As a newcomer, CuPbSbS3 has excellent photoelectric properties, such as high optical absorption coefficient and suitable band gap. The high element abundance makes CuPbSbS3 a potential material for absorbing layers of high-quality thin film solar cells.” is corrected as “CuPbSbS3 has excellent photoelectric properties, such as high element abundance and optical absorption coefficient, and suitable band gap, which makes CuPbSbS3 a potential material for absorbing layers of high-quality thin film solar cells.”.  

Abstract: “CuPbSbS3 films were prepared with butyldithiocarbamic acid (BDCA) solution and deposited on stable substrates such as molybdenum (Mo) or fluorine-doped tin oxide (FTO), the preparation process…… ” is corrected as “CuPbSbS3 films deposited on a stable substrate of molybdenum (Mo) were prepared with butyldithiocarbamic acid (BDCA) solution, and the preparation process…… ”.

Line 28: “…… researchers are widely looking for solar cells with a wide range of absorbing layer materials. Therefore, Cu2ZnSnS4 (CZTS), ….” is corrected as “…… absorbing layer materials as Cu2ZnSnS4 (CZTS), …… ”.

Line 31-32: “ …phase….is…. ” is corrected as “…phases….are…. ”. 

Line 53-58: Adjust the order of sentences to make the meaning clearer: Due to the large number of coordination bonds to be paired on the S atom in butyldithiocarbamic acid (BDCA) solution, it can chelate metal cations very effectively, and form a stable gel, finally precipitate the corresponding metal sulfides through high temperature heating [8,9]. The above BDCA solution method has achieved great success in the preparation of other photovoltaic absorption materials and fully verified the feasibility [8,10]. In this regard, we try to use BDCA solution to prepare the precursor solution.    

Line 110: “ were ” is corrected as “ was ”.

Line 113-114: “a Zennium (Zahner, Zennium) electrochemical workstation ” is corrected as “an electrochemical workstation (Zahner, Zennium)”.

Line 128-129: “In order to explore the formation of substances in this reaction process, it is convenient for us to determine the drying and annealing temperatures of the experiment. So TGA and DSC…… ” is corrected as “In order to explore the formation process of substances and afterwards determine the drying and annealing temperatures of the sintering procedure, TGA and DSC……”. 

Line 133: “…which is because at this temperature, … ” is corrected as “…owing to that… ”.  

Line 151-152: “and the drying temperature and annealing temperature of the final film preparation are 70 °C and 275 °C.” is corrected as “and set the drying and annealing temperatures of the final film preparation to 70 °C and 275 °C accordingly.”.

Line 161, 167, 168, 171, 172, 174, 179 and 181: The grammatical revision of the “be” verb.

Line 183: “pattern shows ” is corrected as “patterns show ”.  

Line 190-191: “Moreover, SEM test also found that this particle only appeared on 1L film, so this particle has little content on the film.” is corrected as “Moreover, it is also found from SEM images that the Sb2S3 particles only appear on 1L film, meaning little content of Sb2S3 secondary phase in the film.”.

Line 205: “…and PbS are a p-type semiconductor light-absorbing material similar to… ” is corrected as “…and PbS are p-type semiconductors and light-absorbing materials similar to…”.

Line 211-213: “CuPbSbS3 is a quaternary compound with complex components, and its theoretical direct band gap is about 1.41 eV [14]. However, due to…” is corrected as “And the calculated band gap of CuPbSbS3 is 2.0 eV which is different from its theoretical direct band gap of 1.41 eV [14]. In fact, due to… ”.

Line 218: “Cu2S (2.42 eV) ” is corrected as “Cu2O (2.2-2.48 eV)”.

Line 220: the Repeated statements need to be simplified, “…the film [16-19]. In this work, we found the tiny secondary phase of p-type Cu2O (2.2-2.48 eV), PbS and Sb2S3 semiconductors in CuPbSbS3 thin film, which will result in a certain change in the band gap value,and we can clearly see the absorption edges of the film in the range of 600-700 nm and 700-800 nm [20]. The band gap value is mainly 2.0 eV indicates that this material can be applied to…” is corrected as “…the film [16-20]. The wide band gap value of 2.0 eV indicates that CuPbSbS3 can be applied to…”

Conclusions: “The emergence of these p-type semiconductors has resulted in the band gap of CuPbSbS3 films being dominated by 2.0 eV, which can be…” is corrected as “The emergence of these p-type semiconductors results in a band gap of 2.0 eV in CuPbSbS3, which can be….”

Sincerely

Limei Lin

Reviewer 2 Report

Comments and Suggestions for Authors

Authors answered suitably for all questions. Hence the Ms is accepted for publication. 

Author Response

Thank you for your kind approval.